# The Differential Effect of Carbon Dots on Gene Expression and DNA Methylation of Human Embryonic Lung Fibroblasts as a Function of Surface Charge and Dose

**DOI:** 10.3390/ijms21134763

**Published:** 2020-07-04

**Authors:** Michal Sima, Kristyna Vrbova, Tana Zavodna, Katerina Honkova, Irena Chvojkova, Antonin Ambroz, Jiri Klema, Andrea Rossnerova, Katerina Polakova, Tomas Malina, Jan Belza, Jan Topinka, Pavel Rossner

**Affiliations:** 1Department of Nanotoxicology and Molecular Epidemiology, Institute of Experimental Medicine of the Czech Academy of Sciences, 14220 Prague, Czech Republic; michal.sima@iem.cas.cz (M.S.); kristyna.vrbova@iem.cas.cz (K.V.); antonin.ambroz@iem.cas.cz (A.A.); 2Department of Genetic Toxicology and Epigenetics, Institute of Experimental Medicine of the Czech Academy of Sciences, 14220 Prague, Czech Republic; tana.brzicova@iem.cas.cz (T.Z.); katerina.honkova@iem.cas.cz (K.H.); irena.chvojkova@iem.cas.cz (I.C.); andrea.rossnerova@iem.cas.cz (A.R.); jan.topinka@iem.cas.cz (J.T.); 3Department of Computer Science, Czech Technical University in Prague, 12135 Prague, Czech Republic; klema@fel.cvut.cz; 4Regional Centre of Advanced Technologies and Materials, Faculty of Science, Palacký University Olomouc, 77146 Olomouc, Czech Republic; katerina.polakova@upol.cz (K.P.); tomas.malina@upol.cz (T.M.); jan.belza01@upol.cz (J.B.); 5Department of Physical Chemistry, Faculty of Science, Palacký University Olomouc, 77146 Olomouc, Czech Republic

**Keywords:** carbon dots, surface charge, human lung fibroblasts, gene expression, DNA methylation

## Abstract

This study presents a toxicological evaluation of two types of carbon dots (CD), similar in size (<10 nm) but differing in surface charge. Whole-genome mRNA and miRNA expression (RNAseq), as well as gene-specific DNA methylation changes, were analyzed in human embryonic lung fibroblasts (HEL 12469) after 4 h and 24 h exposure to concentrations of 10 and 50 µg/mL (for positive charged CD; pCD) or 10 and 100 µg/mL (for negative charged CD, nCD). The results showed a distinct response for the tested nanomaterials (NMs). The exposure to pCD induced the expression of a substantially lower number of mRNAs than those to nCD, with few commonly differentially expressed genes between the two CDs. For both CDs, the number of deregulated mRNAs increased with the dose and exposure time. The pathway analysis revealed a deregulation of processes associated with immune response, tumorigenesis and cell cycle regulation, after exposure to pCD. For nCD treatment, pathways relating to cell proliferation, apoptosis, oxidative stress, gene expression, and cycle regulation were detected. The expression of miRNAs followed a similar pattern: more pronounced changes after nCD exposure and few commonly differentially expressed miRNAs between the two CDs. For both CDs the pathway analysis based on miRNA-mRNA interactions, showed a deregulation of cancer-related pathways, immune processes and processes involved in extracellular matrix interactions. DNA methylation was not affected by exposure to any of the two CDs. In summary, although the tested CDs induced distinct responses on the level of mRNA and miRNA expression, pathway analyses revealed a potential common biological impact of both NMs independent of their surface charge.

## 1. Introduction

Carbon dots (CDs) belong to a relatively new class of nanomaterials with a size smaller than 10 nm but with a large surface area. CDs were first isolated from crude soot in 2004 [1], after which many similar carbon substances were discovered. Due to CDs’ high biocompatibility, photostability, and photoluminescence, they have been used in biomedicine for bioimaging or drug delivery (reviewed by Duran et al. [2]). It has also been shown that the CDs’ surface area allows ligand binding, influencing the behavior of the material, such as trafficking in cells, cellular uptake and cytotoxicity (reviewed by Zhao et al. [3]).

The toxicity of carbon dots has so far been mostly evaluated using the detection of ROS production, cell viability, and cell cycle changes. Thus, Tao et al. [4] showed no viability changes in human kidney embryonic T293T cells, exposed to three different carbon nanoparticles. Other studies have indicated that the cytotoxicity of CDs might be connected with the surface passivation type and not to the CDs themselves [5,6]. Havrdova et al. [7] tested the cytotoxicity of CDs derived from candle soot with three divergent surface functionalization: pristine with negative charge, polyethylene glycol with neutral charge, and polyethyleneimine with positive charge. The effects in in vitro experiments on the standard mouse fibroblasts (NIH/3T3) also varied based on the CD surface charges. Nanoparticles differently influenced the cell cycle, cell morphology, intracellular trafficking, or ROS production. Neutrally charged CDs were the least toxic. The results confirmed that the cytotoxicity was caused by the type of surface modification [7].

The cytotoxic effects of the nanoparticles were also shown to be cell specific. Hsu et al. [8], exposed three various human cancer and two normal control cell lines to carbon dots. A high viability decrease was only observed in the cancer cells and did not appear in the controls. Similarly, high toxicity was observed in human hepatocellular carcinoma cells (HepG2), but not in normal liver cells (FL83B) treated with fluorescent carbon nanodots [9].

In contrast to toxicity markers studied for various types of carbon dots and cell lines, the research focused on gene expression changes due to CDs exposure is sparse. CDs affected the expression of genes encoding antioxidant enzymes via scavenging reactive oxygen species, in rodent mesenchymal stem cells [10]. Li et al. [11] reported expression changes in genes associated with cell cycle progression, cytoskeleton maintenance, or DNA repair in human esophageal epithelial cells exposed to CDs. In human umbilical cord cells and fibroblasts derived from neonatal foreskin, the genes associated with proangiogenic factors were deregulated after exposure to CDs prepared from citric acid and urea [12]. Next generation sequencing of whole genome gene expression changes, after exposure to carbon dots, have so far only been studied in plants. The deregulation of genes responsible for oxidative stress, photosynthetic processes, and the osmotic potential, was observed in *Arabidopsis thaliana* treated with 3 nm CDs [13].

To date, three studies describing the utilization of the positive charged quaternized CDs (pCD) have been published. Maiti et al. [14] hybridized pCDs with DNA and bound these structures to histones, which enabled their sensitive detection by fluorescence monitoring. For the first time, Datta et al. [15] imported these CDs in the cell nucleus of mouse fibroblasts (NIH/3T3) and showed no cell viability changes after using the concentration necessary for cell imaging. Malina et al. [16] used pCDs as a novel stem cell tracking probe in in vivo experiments with transplanted human mesenchymal stromal cells. They described high biocompatibility by studying in vitro toxicity. Based on the results of ROS production, cell viability, cell cycle changes, and DNA breaks assessed by comet assay, pCDs in concentration applicable for imaging, have not been found to cause any negative effects on the native function and behavior of the stromal cells [16].

In this study, we compared the cytotoxic effects, as well as whole genome gene expression changes and gene-specific DNA methylation, in human embryonic pulmonary fibroblasts (HEL 12469) exposed to two types of carbon dots: (1) quaternized carbon dots (pCDs) with positive charge [17] and (2) carbon dots with negative charge (nCDs) [18]. We further identified mRNA targets of miRNAs found to be deregulated in our study and performed an analysis of pathways deregulated by CDs exposure.

## 2. Results

### 2.1. Physico-Chemical Properties of CDs

As pCDs have quaternized a group on the surface originating from betaine, the particles exhibited positive zeta-potential with a peak at 40 mV (Figure 1A). In contrast, the particles prepared from citric acid and urea have typically carboxylic and amidic groups on the surface, which is reflected in the strong negative zeta-potential (Figure 1A). Therefore, the tested particles were denoted as positive CDs (pCDs), or negative (CDs), respectively. The fluorescent excitation-emission maps for both of the prepared CDs are presented in Figure 1B,C. The most dominant fluorescent emission of pCDs was in the blue region of the spectra. The nCDs exhibited an excitation dependent emission, originated from the different types of nitrogen, in the core of the particles as previously reported [18]. Both particles have similar size (diameter of approx. 3 nm as detected by transmission electron microscop; hydrodynamic diameter 5–6 nm as illustrated by dynamic light scattering) and globular (Appendix A).

According to the Fourier transformed infrared spectroscopy results (FT-IR, Figure 1D), both of the prepared particles exhibited a strong vibration at 1600–1650 cm^−1^, corresponding to C=C vibration typical for aromatic regions of the particles (sp^2^ core of the particles). The pCDs had very dominant peaks at 2930 cm^−1^ and 1450 cm^−1^, characteristic for C–H stretching and bending mode reflecting the presence of CH_3_ groups of quaternary ammonium [19]. The FT-IR spectrum of nCDs had peaked at 1105 cm^−1^, a stretching mode of C–O bond. The presence of C=O bond in nCDs was reflected in an extended shoulder (1710 cm^−1^) of C=C vibration. The x-ray photoelectron survey spectra (XPS) of the prepared samples (Figure 1E,F) show the presence of carbon, oxygen and nitrogen in both samples (for details, see Appendix A). The chlorine in pCDs reflects the presence of a counter ion to the surface quaternary ammonium group. The high resolution C 1s XPS spectra of both samples revealed the presence of C=C, C-O/C-N, C=O and O-C=O bonds (Figure 1E,F, Appendix A). According to the expected surface functional groups, the pCDs had a more dominant C-N peak. On the other hand, the nCDs had more dominant carboxylic and carbonyl groups.

### 2.2. Cell Viability after pCD and nCD Exposure

pCD treatment caused a decrease in cell viability (below 95% when compared with the control) at concentrations ≥ 100 µg/mL, while for nCDs, even at the highest concentration tested (500 µg/mL), *the viability remained over 95*% (Figure 2). Furthermore, at a concentration of 50 µg/mL, both pCD and nCD were internalized in cellular vesicles as indicated by fluorescence signal from confocal microscopy in four different layers of z plane (Appendix A). Based on these results we selected the following concentrations of CDs for mRNA, miRNA and DNA methylation experiments: 10 and 50 µg/mL; and 10 and 100 µg/mL, for pCD and nCD tests, respectively.

### 2.3. Expression of mRNAs Induced by pCD and nCD NPs Treatment

The incubation of HEL12469 cells with pCD affected the expression of a lower number of mRNAs than those exposed to nCD (Appendix A). For pCD, we observed a differential expression of 3, 61, 8 and 727 mRNAs, after incubation with 10 µg/mL for 4 h (4h10), 50 µg/mL for 4 h (4h50), 10 µg/mL for 24 h (24h10) and 50 µg/mL for 24 h (24h50), respectively (Figure 3A, Appendix A). Most of these deregulated mRNAs were unique for individual exposure conditions. There were few commonly expressed mRNAs, of which only one mRNA was common for exposure to 24h10, 24h50 and 4h50 of pCD (*DCBLD2*, encoding discoidin, CUB and LCCL domain-containing protein 2) (Appendix A). The number of downregulated mRNAs was higher than those that were upregulated (2, 35, 7 and 432; 1, 26, 1 and 295, for downregulated and upregulated mRNAs after exposure to 4h10, 4h50, 24h10 and 24h50 pCD, respectively) (Appendix A). Hierarchical clustering of top-ranked differentially expressed mRNAs in the exposed samples when compared with the controls, is shown in Figure 4. Selected significantly deregulated mRNAs with |log2FC| > 1 and adjusted *p*-value <0.05 are graphically shown in volcano plots (Figure 5A–D). The differentially expressed genes include e.g., those encoding Rho GTPase activating protein 26 (*ARHGAP26*), tripartite motif-containing protein 16 (*TRIM16L*) and CCAAT/enhancer-binding protein delta (*CEBPD*) (4h10), protein sprouty homolog 4 (*SPRY4*], stanniocalcin-1 (*STC1*], prostaglandin H synthase (*PTGS2*), putative survival-related protein (*CASZ1*), and ATP-binding cassette sub-family A member 3 (*ABCA3*) (4h50), leukotriene B4 receptor 2 (*LTB4R2*), nuclear receptor coactivator 7 (*NCOA7*) (24h10), insulin induced gene 1 (*INSIG1*), transforming growth factor, β-induced (*TGFBI*), cell division cycle-associated protein 3 (*CDCA3*) and cell-division cycle protein 20 (*CDC20*) (24h50).

The incubation of HEL12469 cells with nCD resulted in the deregulation of 147, 2463, 383 and 2266 mRNAs [for 10 µg/mL for 4 h (4h10), 100 µg/mL for 4 h (4h100), 10 µg/mL for 24 h (24h10) and 100 µg/mL for 24 h (24h100), respectively (Figure 3B, Appendix A)]. In contrast to pCD, for nCD we found more commonly deregulated mRNAs, of which twelve were common for all exposure conditions (Figure 3B, Appendix A). The numbers of downregulated mRNAs were as follows: 78, 912, 203 and 977 for 4h10, 4h100, 24h10 and 24h100 exposure, respectively. For upregulated mRNAs, we detected 69, 1551, 180 and 1289 differentially expressed genes after exposure to 4h10, 4h100, 24h10 and 24h100, respectively (Appendix A). Hierarchical clustering of top-ranked differentially expressed genes detected in the treated samples vs. the controls for individual exposure conditions, is reported in Figure 6. Selected significantly differentially expressed mRNAs with log2FC >1 and adjusted *p*-value >0.05 is shown in Figure 7. The following notable deregulated transcripts were detected: ERBB receptor feedback inhibitor 1 (*ERRFI1*), leukemia inhibitory factor (*LIF*), dual specificity protein phosphatase 1 (*DUSP1*), interleukin 11 (*IL11*) (4h10, 4h100), G2 and S phase-expressed protein 1 (*GTSE1*), cell-division cycle protein 20 (*CDC20*), LBH regulator of Wnt signaling pathway (*LBH*) (24h10), cell division cycle-associated protein 3 (*CDCA3*), G2 and S phase-expressed protein 1 (*GTSE1*), ribonucleoside-diphosphate reductase subunit M2 (*RRM2*), endothelial PAS domain-containing protein 1 (*EPAS1*) and cyclin-dependent kinase 2 (*CDK2*)(24h100).

We further compared the differences between mRNAs affected by exposure to individual NPs. After 4 h incubation, most of the deregulated mRNAs were unique for the tested NPs however we found a total of 25 mRNAs commonly differentially expressed after exposure to both materials, regardless of their concentration. One of these mRNAs (*ARHGAP26*, encoding Rho GTPase activating protein 26) was common for all exposure conditions (Figure 8A, Appendix A). A dose of 10 µg/mL, that was common for both NPs, affected the differential expression of one common gene. Most of the commonly deregulated mRNAs were downregulated (19 transcripts); upregulation was observed for the differential expression of six genes. The longer incubation resulted in the deregulation of a higher number of common transcripts; 175 mRNAs differentially expressed after exposure to both NPs, regardless of concentration. Two genes were commonly differentially expressed for all tested conditions (*DCBLD2*, encoding discoidin, CUB and LCCL domain containing two proteins; *GFRA1*, encoding GDNF family receptor α 1 protein); one gene after exposure to a dose of 10 µg/mL of both materials (IGFBP3, encoding insulin-like growth factor binding protein 3) (Figure 8B, Appendix A). Similarly to short exposure, 24 h incubation was associated with the downregulation of a greater number of mRNAs, than with the gene upregulation (118 vs. 50 mRNAs).

In agreement with the findings for deregulated mRNAs, the functional enrichment analysis revealed differences between the number of deregulated pathways after exposure to pCD and nCD. For pCD, significantly deregulated pathways were identified only after 24h50 exposure (Table 1). These pathways were related to cell cycle and its regulation (e.g., Cell Cycle, Cell Cycle Checkpoints, G1/S Transition, M Phase) as well as cholesterol biosynthesis, which is also related to cell cycle progression.

Exposure to nCD resulted in the deregulation of a higher number of pathways. The 100 µg/mL concentration applied for 4 h, affected protein translation (Eukaryotic Translation Elongation, Eukaryotic Translation Initiation, Eukaryotic Translation Termination), Gene expression, and Pathways in cancer. Similarly to pCD, the 24 h treatment with nCD was associated with processes related to cell cycle regulation, additionally possibly linked with processes of carcinogenesis for the lower dose (p53 signaling pathway), and protein expression and DNA repair for the higher dose (Table 2). The overview of deregulated pathways is graphically illustrated in Figure 9, details are provided in Appendix A.

### 2.4. The Expression of miRNAs Affected by Treatment with pCD and nCD NPs

The short, 4 h exposure of HEL12469 cells to a lower dose of pCD NPs, had no effect on miRNA expression. For other tested doses and exposure periods, deregulated miRNAs were found: 3, 31 and 76 miRNAs, after incubation with 4h50, 24h10 and 24h50, respectively (Figure 10A, Appendix A). We observed a very small overlap of deregulated miRNAs between both exposure periods: only three molecules, regardless of concentration, were commonly expressed after exposure for 4 h and 24 h (Appendix A).

Incubation of the cells with nCD NPs affected the higher number of miRNAs when compared with pCD NPs: 25, 147, 77 and 220 miRNAs, were differentially expressed after exposure to 4h10, 4h100, 24h10 and 24h100, respectively (Figure 10B, Appendix A). Although 142 miRNAs were unique, we found a total of 28 miRNA commonly deregulated for both time periods, of which 19 were common for both concentrations and exposure times. Interestingly, we observed 82 miRNAs to be commonly expressed after exposure to 100 µg/mL for both time periods (Appendix A).

We then focused on the differences in miRNA expression induced by both the tested nanomaterials, after individual exposure periods. As expected, after the short treatment, we found no miRNAs commonly deregulated for both materials (Figure 11A, Appendix A). The longer incubation resulted in a common deregulation of a total of 20 miRNAs, of which two were common for all exposure conditions. Only two deregulated miRNA genes were detected for the treatment with the common concentration of 10 µg/mL (Figure 11B, Appendix A).

The number of differentially expressed validated miRNA targets detected in our study, out of the total number of targets identified using a database search, for miRNAs significantly deregulated after exposure to pCD and nCD NPs are reported in Appendix A. In agreement with the observations described above, the number of such targets was low for pCD NPs treatment; significant results were only observed for the 24h50 exposure. In contrast, for all nCD treatment conditions, differentially expressed validated targets were identified; their numbers generally increased with the dose and time of exposure.

We further performed the functional enrichment analysis of significantly deregulated miRNAs based on experimentally validated miRNA-mRNA interactions, with the aim to identify pathways affected by exposure to pCD and nCD NPs. In general, after 24 h exposure, we observed a great overlap of deregulated pathways (Table 3 and Table 4). Both CDs impacted cancer-related pathways (e.g., Colorectal cancer, Pancreatic cancer, Endometrial cancer), as well as other pathways associated with tumorigenesis (e.g., Wnt signaling pathway, Ras signaling pathway, ErbB signaling pathway), immune response-related processes (T cell receptor signaling pathway, Fc gamma R-mediated phagocytosis) and cell-extracellular matrix (ECM) interactions (Mucin type O-Glycan biosynthesis). While no significant results were found after 4 h pCD treatment, nCD caused the deregulation of pathways involved in cell-extracellular matrix (ECM) interactions (Mucin type O-Glycan biosynthesis, ECM-receptor interaction). These NPs further affected fatty acid biosynthesis and apoptosis. The overview of pathways identified based on miRNA-mRNA interactions is reported in Figure 12, details are provided in Appendix A.

### 2.5. DNA Methylation Changes after Treatment with pCD and nCD NPs

An investigation of gene specific DNA methylation changes was performed with the aim to identify differentially methylated CpG sites and regions. However, we did not find any significant results for either exposure condition, suggesting that the tested NPs have any effect on DNA methylation under the selected treatment conditions (NPs concentration and/or time of exposure) (data not shown).

## 3. Discussion

Our results showed, for the first time in human cells, that treatment with CDs gave rise to the deregulation of mRNA and miRNA gene expression and that the resulting deregulated pathways were shared for the different conditions we tested. Nevertheless, we saw a trend for a dose/exposure-time response curve, with higher doses and longer exposure time, showing more deregulation. Furthermore, we found that the CD’s surface charge is important, with the positively charged pCD giving rise to less deregulation than the negatively charged nCD nanoparticles. In terms of epigenetic changes, we saw a deregulation of miRNA pathways, but no DNA methylation changes. For our experiments we used human embryonic lung fibroblasts as a model system. Although fibroblasts do not represent a primary target for particles entering the pulmonary system, they become exposed in damaged areas of the lungs, e.g., after exposure to various pollutants, including nanoparticles [20,21]. In such tissues, fibroblasts proliferate and consequently play a major role in the response to external stimuli. Additionally, due to their small size, CDs could easily penetrate through the epithelial cells and affect other cells, including fibroblasts, even if the lung epithelium is intact [22].

Carbon dots have potential to be used as a labeling agent not only in regenerative medicine [23], but also for targeted imaging and gene delivery in cancer therapy, including lung tumors [24,25]. For such treatments, it is paramount that healthy tissues are not negatively impacted by the introduction of foreign particles. The studies analyzing the negative biological effects of CDs have usually focused on the detection of cytotoxicity, cell cycle alterations or ROS production. The authors mostly noted the very low cell-specific and surface passivation-dependent toxicity of these NMs (e.g., [4,5,6,7,8,9]). However, in a study performed in a plant model *Arabidopsis thaliana* (L.), the negative impacts of CDs exposure on root elongation were observed for higher tested doses, suggesting phytotoxicity in this organism [13]. In that study, a comprehensive genomics analysis using next generation sequencing (NGS) was conducted. The expression of more than 600 genes in the roots and shoots were upregulated, including those involved in cellular response to phosphate starvation, stimulus response, and UDP-glycosyltransferase activity. On the contrary, the expression of more than 500 genes, mainly those responsible for chloroplast functions and structure, was downregulated. However, no comparable assessment of CDs effects on genomics parameters was conducted in animal systems.

To date, very limited information on the potential toxicity of pCD and nCD used in this study is available. In mouse fibroblasts NIH/3T3 exposed to pCD, the viability detected using the MTT assay was not affected after treatment with concentrations relevant for cell imaging [15]. Malina et al. [16] exposed human mesenchymal stromal cells to pCD. By the detection of ROS generation, analyses of cell viability, cell cycle changes, and DNA fragmentation by comet assay, they discovered that these carbon nanoparticles are biocompatible, even at a very high dose (400 µg/mL) and did not cause a negative effect on the behavior or native function of the cells at a dose of 100 µg/mL. Thus, in these cell models pCD may represent a suitable material for biomedical applications. However, no such evidence was reported for other cell types. Moreover, to the best of our knowledge, no toxicological studies were performed with nCD.

In this study, treatment with a positively charged pCD resulted in the reduction of cell viability, already detectable at a concentration of 100 µg/mL. In contrast, exposure to negatively charged nCD had no measurable adverse impact on viability, even at a dose of 500 µg/mL. Similar data showing surface charge-dependent toxicity were obtained by Havrdova et al. [7] for another group of CDs. In agreement with our cell viability data, the changes of mRNA and miRNA expression evaluated as the number of deregulated transcripts were more pronounced in samples treated with nCD, than in those exposed to pCD. This result was obtained already at a dose of 10 µg/mL when no reduction of cell viability was detected. We speculate that pCD had inhibitory effects on gene expression even at the lower tested concentration that was not detectable by the live-dead analysis. Such response may have been manifested by lower numbers of deregulated genes induced by pCD.

For mRNA expression, we observed some notable unique differentially expressed molecules for individual exposure periods and doses of CDs. Although these changes suggest the possible direction of cellular impacts of the tested CDs, it should be noted that the biological significance of modulations based on the deregulation of single genes is limited and that pathway analyses provide better overall demonstration of impacts of the investigated NMs on the organism.

Thus, after 4h10 exposure to pCD, the expression of *ARHGAP26* encoding a protein participating in endocytosis and associated with cancer was upregulated, and *TRIM16L* involved in cell growth, differentiation and pathogenesis, as well as *CEBPD* encoding the protein involved in immune and inflammatory responses were downregulated, suggesting the potential impacts of these CDs on immunity, surfactant production [26] and lung carcinogenesis [27]. The effects modulated by the 4h50 exposure to pCD were mostly comparable. We observed a reduced expression of *STC1*, that encodes stanniocalcin 1, a protein reported to functionally participate in the development of some cancers, including lung cancer [28]. The expression of *SPRY4* that has been shown to inhibit cell growth and migration in non-small lung cancer [23] was also downregulated. The upregulation of *ABCA3*, encoding multi-membrane spanning protein, that plays a critical role in pulmonary surfactant homeostasis [24] was detected. Finally, the potential anti-inflammatory role of pCD exposure was suggested by a decreased expression of *PTGS2*.

After 24 h of treatment, pCD reduced the expression of genes related to inflammation (*CCL2*, *LTB4R2*). In addition, the expression of *TGFBI* encoding a protein that plays a role in cell-collagen interactions was upregulated. The loss of expression of this gene was reported in several cancers, including lung cancer [25]. The expression of genes encoding cell division cycle-associated proteins (*CDCA3*, *CDC20*) was downregulated. The modulation of their expression was associated with lung cancer [29,30]. Finally, the genes encoding proteins involved in mitochondria functions and oxidative phosphorylation (*MT-ATP6*, *MT-ATP8*, *MT-CO2*, *MT-CO3* or *MT-CYB*), were affected by these CDs. Overall, the data indicate the potential inhibitory role of pCD on cell cycle progression, as well as an association with oxidative stress, but also the possible induction of apoptotic response [31].

Four hours of exposure to nCD downregulated the expression of: *ERRFI1* whose protein product regulates DNA damage response and promotes DNA repair [32]; *LIF* that is involved e.g., in inflammatory response [33]; and *DUSP1* that plays a role in the response to stress and negatively regulates cellular proliferation [34]. We further detected the deregulation of genes encoding proteins interacting with p53 tumor suppressor (*TP53BP1*, *MDM2*) or cyclins and other cell cycle regulation genes (e.g., *CCNB1*, *CCNF*, *CCNI*, *CCNK*, *CCNL1*, *CCNT2* or genes encoding cyclin-dependent kinases). Thus, 4 h exposure to nCD is potentially linked with the disturbance of DNA damage response and carcinogenesis [35].

The expression of *CDC20* was downregulated by the 24h10 treatment with nCD. These NPs further reduced the expression of *GTSE1,* whose protein product binds p53 and thus blocks apoptosis. Its expression was observed in radiation-induced lung fibrosis [36]. The exposure to 24h50 nCD was associated with the downregulation of *GTSE1* as well as *CDCA3* and expression. Similar results were observed for *RRM2* and *CDK2*. The protein encoded by *RRM2* is linked with the synthesis of deoxyribonucleotides from ribonucleotides, and its expression is associated with poor prognosis in non-small cell lung cancer patients [37]. CDK2, an important regulator of cell cycle, was consistently found to be overexpressed in human cancers, including lung cancer [38]. *DDB2* encoding DNA damage-response protein, genes encoding subunits of mitochondrial ATP synthase, collagen, as well as oncogenes and tumor suppressor genes, including *TP53* were also modulated by nCD exposure. Thus, apart from the possible impacts on oxidative stress and apoptosis described above, these CDs might affect collagen deposition and therefore contribute to lung fibrosis [39]. The link of nCD exposure with tumorigenesis is not consistent and final conclusions cannot be drawn based on the obtained data.

As already suggested, although the numbers of differentially expressed genes and pathways differed for the tested CDs and few genes were commonly expressed, the altered biological functions and processes partly overlapped. This was further confirmed by the analysis of deregulated pathways based on mRNA expression that showed the impact of both CDs on cell cycle regulation. Therefore, our data indicate that both tested CDs share similar mechanisms by which they affect mRNA expression, although the particular doses, exposure times and impacted processes may differ.

To assess the impact of CDs exposure on epigenetic parameters, we focused on miRNA and DNA methylation alterations. While miRNA expression changes may occur during a short time-span, DNA methylation is a slower process with long-lasting consequences. Thus, we expected differences in behavior of these epigenetic markers. As already described, for differential miRNA expression analysis, we detected changes comparable with those found for mRNA. This observation indicates that this parameter readily responds to the presence of CDs in the treated cells. The analysis of pathways related to miRNA expression changes showed a very similar pattern for both CDs. Although the number of pathways deregulated by pCD exposure was substantially lower than that affected by nCD, both nanomaterials were involved in the deregulation of various tumorigenesis-related pathways, extracellular matrix interactions and immune response. In addition, exposure to nCD affected apoptosis.

The analysis of gene-specific DNA methylation did not show an effect of either type of CDs on this epigenetic parameter. Although this is the first study in which such analysis was performed, we believe that short exposure times are responsible for the negative observations. In a recent study, persistent global DNA methylation changes were found in zebrafish exposed to various nano-graphene quantum dots for 7 days [40]. Although the experimental model and the method of DNA methylation detection were different than in our study, the longer exposure period was most likely the reason for the positive results observed by these authors. Our results suggest that, from the long-term perspective, short-term exposures are not deleterious for the organism. The changes detected on an mRNA level are probably limited to a short period of time. Thus, to elucidate the potential impact of pCDs and nCDs on gene-specific DNA methylation changes in HEL 12469 cells, a different time exposure scheme involving prolonged exposure periods will be necessary.

In summary, we described the whole genome gene expression changes in human embryonic lung fibroblasts after exposure to two types of carbon dots—positively charged pCD and negatively charged nCD. Despite differential cellular response, most likely related to distinct surface charge, manifested by the deregulation of a greater number of mRNAs, miRNAs and associated pathways after exposure to nCD, both types of CDs seem to have similar biological impacts. These are associated with the potential changes of immune response and cell cycle-related processes. Nevertheless, pCD, especially at low doses, appear to show a more learned profile. However, our data should be interpreted with caution, as they mostly reflect processes on the transcription level without corresponding confirmation of the expression of translated proteins. Consequently, for future biomedical applications of these CDs more tests, preferentially in animal models, should be carried out. These experiments should include not only genomic analyses, but also the detection of relevant phenotypic markers.

## 4. Materials and Methods

### 4.1. Carbon Dots and Their Characterization

The materials were obtained according to our previous protocols [17,18]. The positive charged carbon dots were prepared by solid-state carbonization of betaine hydrochloride and tris (hydroxymethyl) aminomethane (TRIS). The negative charged CDs were produced by the solvo-thermal method; the autoclave decomposition of citric acid and urea in dimethylformamide (DMF). The only modification was in the preparation of nCDs, mainly in the purification protocol. The prepared particles were not separated by ionex-chromatography as previously reported, but repeatedly washed with water over a 3 kDa ultrafiltration membrane (Millipore ultrafiltration tube). The unreacted precursor and DMF passed through the filter. The purified nCDs stayed in the ultrafiltration inset. The particles were subsequently freeze-dried and dissolved, before use at required concentrations.

Fluorescence spectroscopy was performed on a FLS980 fluorescence spectrometer (Edinburgh Instruments, Livingston, UK) equipped with a R928P photomultiplier (Hamamatsu, Hamamatsu City, Japan) in a thermoelectrically cooled housing, with a 450 W xenon arc lamp as the excitation source for steady-state spectra and an EPL-375 picosecond pulsed diode laser (λem =372 nm; Edinburgh Instruments). Zeta potential (ξ potential) of CDs was measured using Zetasizer NanoZS (Malvern, Worcestershire, UK).

FTIR spectra were measured using Nicolet iS5 FTIR spectrometer (Thermo Fisher Scientific, Waltham, MA, USA) equipped with the Smart Orbit ZnSe ATR technique. Survey and high-resolution C1s XPS spectra were assessed by X-ray photoelectron spectroscopy (XPS) and carried out with a PHI VersaProbeII (Physical Electronics, Chanhassen, MN, USA) spectrometer using an A1 Kalfa source (15 kV, 50 W). The obtained data were evaluated with the MultiPak (Ulvac – PHI, Inc., Kanagawa, Japan) software package and are referred to C 1s peak at 284.80 eV.

Size and shape of the CDs were characterized by transmission electron microscopy (TEM, JEOL 2100 operating at 160 kV, JEOL, Akišima, Tokyo, Japan). Zeta potential (ξ potential) and DLS (dynamic light scattering) of CDs were measured using Zetasizer NanoZS.

The confirmation of pCD and nCD uptake by the cells was performed using a confocal microscope (Olympus, Tokyo, Japan) equipped with laser with 488 nm excitation and detector for emission in the region of 520 to 560 nm. The measurement was done by 60× objective with 1.5× zoom. HEL12469 cells were treated with 50 µg/mL of pCDs and nCDs for 24 h, then the sample was washed with PBS. The measurement was conducted in PBS.

### 4.2. Cell Culture Treament and Viability Assay

HEL 12469 cells were grown at 37 °C in a humidified atmosphere containing 5% CO_2_ in Eagle’s Minimum Essential Medium (EMEM) supplemented with 10% fetal bovine serum, 2 mM glutamine, 1% non-essential amino acids and 0.2% sodium bicarbonate. After reaching 70–80% confluence, the medium was replaced with a complete fresh medium. Prior to exposure, the medium was removed; the cells were then washed and treated with freshly prepared dilutions of only CDs or media (negative controls; NC). Each CD was tested in triplicate.

Cell viability was evaluated using a live/dead analysis. The cells were seeded in 96-well plates (10^4^ cells/well) and cultivated overnight in a culture medium. The cells were then treated with various concentrations (10–500 µg/mL) of pCDs and nCDs. After 24 h, the supernatant was collected and the cells were washed with a phosphate-buffer solution (PBS, 0.1 M, pH 7.4), detached with trypsin (0.25% in ethylenediaminetetraacetic acid (EDTA), Sigma-Aldrich, St. Louis, MO, USA) and resuspended in 150 µl of medium. Every solution added to the cells was collected so that every cell of the sample could be analyzed and not removed during the preparation. The cells were incubated with propidium iodide (PI) (final concentration 10 µg/mL) and calcein-AM (final concentration 50 µM) for 30 min in the dark. The fluorescence signal was then measured on a flow cytometer BD FACSVerse (BD Biosciences, San Jose, CA, USA) using red and green channels (red: ex. 488/em. 700 nm, green: ex. 488/em. 527 nm). The dead cells gave a strong red signal, as PI revealed cells that had lost membrane integrity; whereas the live cells generated a green signal from highly fluorescent calcein, which was catalyzed from non-fluorescent calcein-AM by active intracellular esterases. Mean ± standard deviation (SD) was calculated from the independent experiments. Student t-test was performed using Statistica software (TIBCO, Palo Alto, CA, USA, 2018). Any difference was considered significant and highly significant at *p* ≤ 0.05 and *p* ≤ 0.01, respectively. Two concentrations were selected for gene expression and DNA methylation analyses. Due to toxicity, pCDs were tested at a maximum concentration of 50 µg/mL.

### 4.3. DNA, RNA, and miRNA Extraction

The total RNA and DNA was extracted using AllPrep DNA/RNA/miRNA Universal Kit (Qiagen Manchester Ltd., Manchester, UK). The extraction protocol followed the manufacturers’ instructions with some modifications (purification and precipitation with sodium acetate/ethanol). The concentration of nucleic acids was determined by Nanodrop ND-1000 Spectrophotometer (Thermo Fisher Scientific, Waltham, MA, USA); RNA integrity number (RIN) was assessed by Agilent Bioanalyzer (Agilent Technologies Inc., Santa Clara, CA, USA) with RNA 6000 Nano kit (Agilent Technologies Inc., Santa Clara, CA, USA). RIN values were >9.0 in all samples.

### 4.4. Whole Genome Transcriptome Analysis by Next Generation Sequencing

RNA samples (concentration >210 ng/µL, RNA integrity number >9.0) were used for RNA libraries preparation using [(Poly(A)RNA Selection, Lexogen Sense Total RNA-Seq Library Prep Kit (Lexogen GmbH, Vienna, Austria)]. For sequencing, NextSeq^®^500/550 High Output Kit v2 (75 cycle) and NextSeq 500/550 system (Illumina, Inc., San Diego, CA, USA) were used. The reactions were performed according to the manufacturers’ recommendations.

### 4.5. miRNA Expression Analysis

Small RNA libraries were prepared from 150 ng of total RNA, essentially using Qiaseq miRNA library kit (Qiagen Manchester Ltd., Manchester, UK) as previously described [41]. The kit not only allows the detection of miRNAs, but also other small RNAs, including piRNAs. In our data, most of the results consisted of deregulated miRNAs, with a minor proportion of piRNAs. Thus, the “miRNA expression” reported here, denotes an expression of both miRNAs and piRNAs. Small RNA libraries were validated on a Fragment Analyzer (Agilent Technologies Inc., Santa Clara, CA, USA) with a high sensitivity NGS kit. The sequencing of sRNA libraries was performed on MiSeq system (Illumina, Inc., San Diego, CA, USA) using MiSeq Reagent Kit V3 (Illumina, Inc., San Diego, CA, USA).

### 4.6. Gene Specific DNA Methylation Analysis

Infinium MethylationEPIC BeadChips (Illumina, Inc., San Diego, CA, USA), allowing the interrogation of more than 850,000 CpG loci, dispersed through the whole human genome were used. A total of 500 ng DNA was treated overnight with sodium bisulfite using Zymo EZ DNA Methylation^TM^ Kit (Zymo Research, Irvine, CA, USA) for the conversion of unmethylated cytosines to uraciles, while methylated cytosines remain unchanged. Bisulfite-converted DNA was processed according to the manufacturer’s protocol including the enzymatic fragmentation, precipitation, resuspension and overnight hybridization, followed by washing and BeadChip staining. All chips were scanned by iScan System (Illumina, Inc., San Diego, CA, USA). The methylation status at each CpG site was estimated by measuring the intensity of the pair of methylated and unmethylated probes.

### 4.7. Data Analysis

An NGI-RNAseq pipeline (https://github.com/SciLifeLab/NGI-RNAseq) was used for RNA sequencing data. It pre-processed raw data from FastQ inputs (FastQC, Trim Galore!), aligned the reads (HiSAT2) using reference genome Homo sapiens assembly GRCh37 (hg19), generated gene counts (featureCounts) and performed extensive quality-control on the results (RSeQC, dupRadar, Preseq, edgeR, MultiQC). DESeq2 with default parameter settings was applied to normalize read counts and to identify the differences in gene expression between sample groups. A functional enrichment analysis of differential gene expression data was performed by ToppFun tool, using the feature “Pathway” [42].

Differential miRNA expression count data were generated by QIAseq miRNA Quantification, and the UMI counts were taken to compensate for sequencing bias. The DESeq2-package was used to test for differential expression by the application of negative binomial generalized linear models. A *p* <0.02 was used as the cut-off for statistically significant deregulated miRNAs between sample groups. Heatmaps were used to visualize the magnitude of expression in particular samples. In each experiment, the 20 most significant transcripts reported by DESeq2 were shown. The color corresponds to z-score reached for DESeq2 normalized read count values, i.e., each field visualizes the number of standard deviations by which the normalized read count is above or below the mean value observed for the given transcript in the given experiment (6 samples). Heatmap.2 function from gplots R package was employed for heatmaps generation. The relationship between significance and fold change reported by DESeq2 was shown using volcano scatterplots. All the transcripts (each represented by a single point) were plotted to show the overall distribution. The ggplot2 package was used to generate the plot.

The mRNA-miRNA correlation analysis was performed with the R package miRComb [43]. First, the sets of differentially expressed mRNAs and miRNAs were identified (the detection thresholds were adjusted pval <0.01, FC >1.5 and adjusted pval <0.1, for mRNA and miRNA expression, respectively; the detection method was DESeq2). Correlations between all the mRNA and miRNA candidate pairs were then calculated as previously described [41].

The differential methylation analysis was carried out with the Bioconductor workflow recommended in [44]. In particular, differential methylation was detected with limma package and its empirical Bayes method, suitable for experiments where the number of arrays is small. The minfi package was used for the preprocessing, filtering and quality control, of raw methylation data. Stratified quantile normalization implemented in minfi, was also employed. The DMRcate package eventually served to extract the most differentially methylated regions.

## Figures and Tables

**Figure 1 ijms-21-04763-f001:**
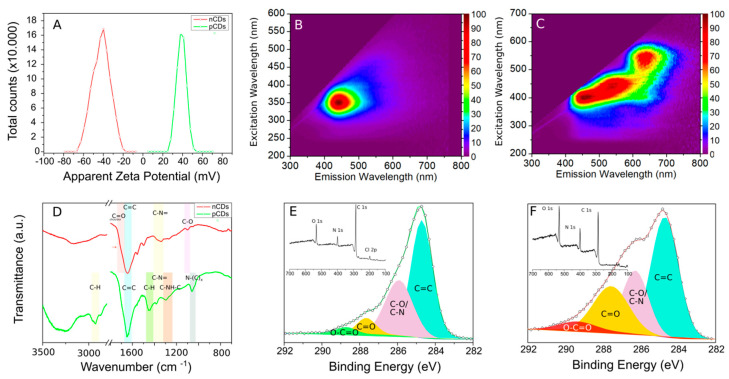
Zeta-potential of the prepared CDs in water at pH 7.0 (**A**); fluorescent excitation-emission map of pCDs (**B**); fluorescent excitation-emission map of nCDs (**C**); FT-IR spectra of pCDs and nCDs with highlighted characteristic vibration bands (**D**); high-resolution C 1s XPS spectrum of pCDs with survey spectrum in the inset (**E**); high-resolution C 1s XPS spectrum of nCDs with survey spectrum in the inset (**F**).

**Figure 2 ijms-21-04763-f002:**
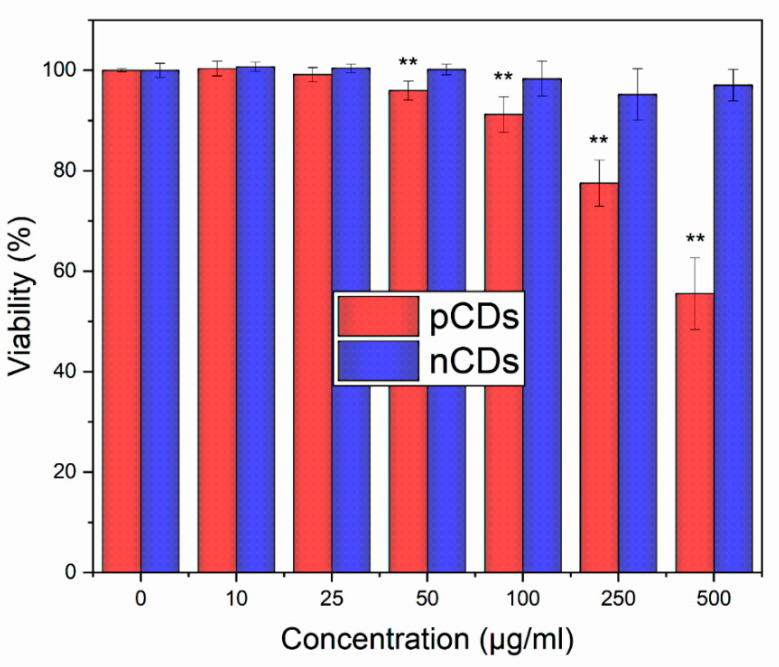
Live-Dead analysis of HEL cells after 24 h treatment with various concentrations of pCDs and nCDs. The results were normalized to the control sample (100% viability). * *p* < 0.05; ** *p* < 0.01.

**Figure 3 ijms-21-04763-f003:**
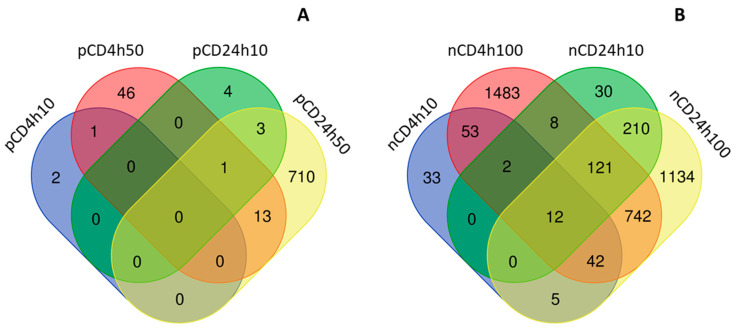
The distribution of common and unique mRNAs differentially expressed after CD NPs exposure; a comparison between doses and exposure times for pCD (**A**) and nCD (**B**).

**Figure 4 ijms-21-04763-f004:**
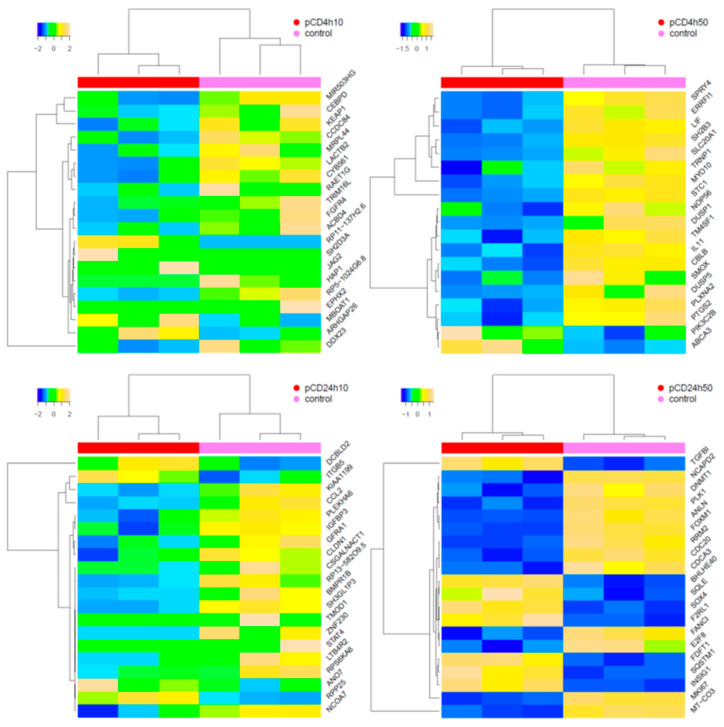
Hierarchical clustering of top-ranked differentially expressed mRNAs in samples exposed to pCD when compared with the controls. The 20 most significantly deregulated transcripts are shown.

**Figure 5 ijms-21-04763-f005:**
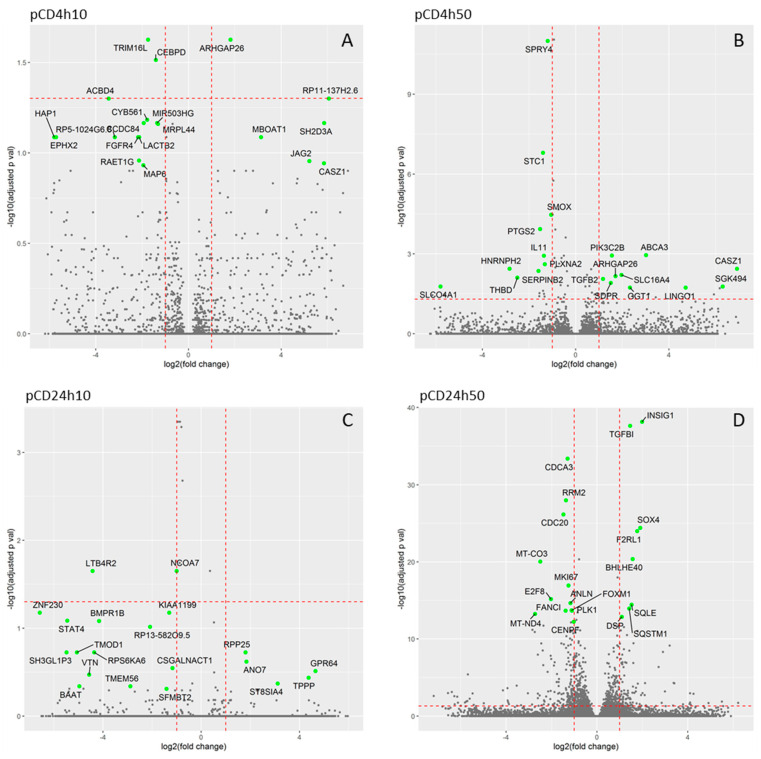
Volcano scatter-plots showing the relationship between significance and fold change of individual mRNA transcripts expression. The horizontal dashed red line highlights the adjusted *p*-value of 0.05, the vertical lines denote the fold changes equal of 0.5 and 2. In each plot the 20 most significant transcripts are identified. (**A**)- pCD4h10, (**B**)- pCD4h50, (**C**)- pCD24h10, (**D**)- pCD24h50.

**Figure 6 ijms-21-04763-f006:**
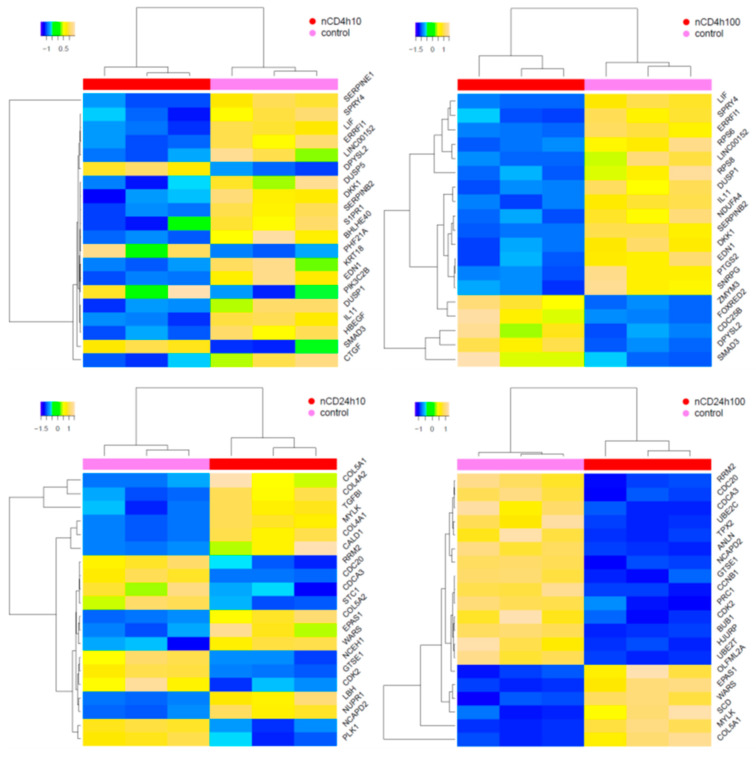
Hierarchical clustering of top-ranked differentially expressed mRNAs in samples exposed to nCD when compared with the controls. The 20 most significantly deregulated transcripts are shown.

**Figure 7 ijms-21-04763-f007:**
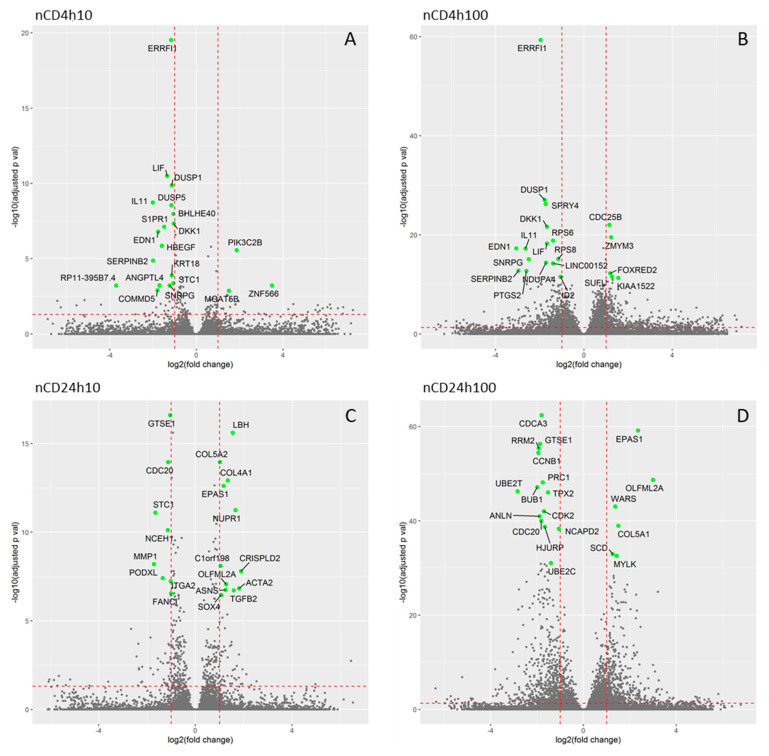
Volcano scatter-plots showing the relationship between significance and fold change of individual mRNA transcripts expression. The horizontal dashed red line highlights the adjusted *p*-value of 0.05, the vertical lines denote the fold changes equal of 0.5 and 2. In each plot the 20 most significant transcripts are identified. (**A**)-nCD4h10, (**B**)-nCD4h100, (**C**)-nCD24h10, (**D**)-pCD24h100.

**Figure 8 ijms-21-04763-f008:**
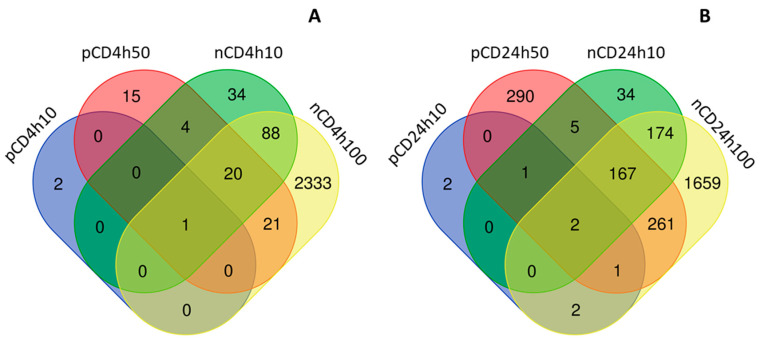
The distribution of common and unique mRNAs differentially expressed after CD NP exposure; a comparison between CDs exposure for 4 h (**A**) versus 24 h (**B**).

**Figure 9 ijms-21-04763-f009:**
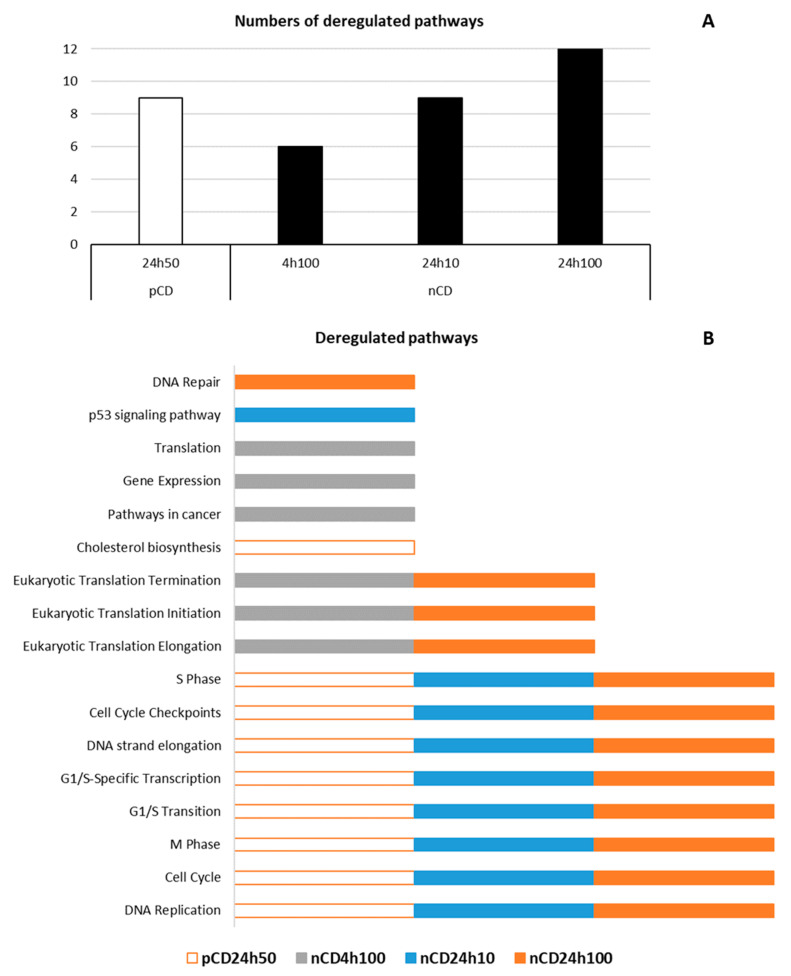
The graphically illustrated overview of deregulated pathways after CD NPs exposure. (**A**) indicates the number of deregulated pathways, meanwhile (**B**) shows in which exposure conditions the selected pathways are deregulated (empty rectangles denote pCD exposure, filled rectangles denote nCD exposure).

**Figure 10 ijms-21-04763-f010:**
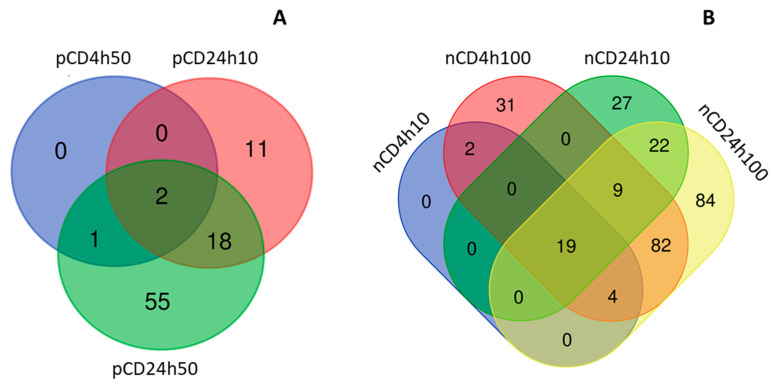
The number of common and unique miRNAs deregulated after CD NPs exposure; a comparison between doses (**A**) and exposure times (**B**) for individual CDs.

**Figure 11 ijms-21-04763-f011:**
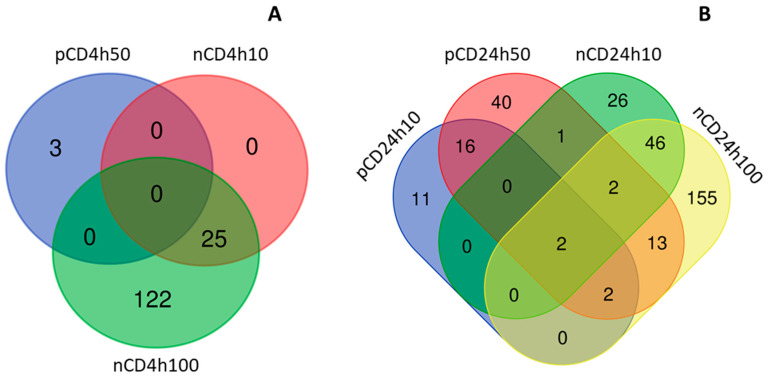
The numbers of common and unique miRNAs deregulated after CD NPs exposure; a comparison between CDs for individual doses (**A**) and exposure times (**B**).

**Figure 12 ijms-21-04763-f012:**
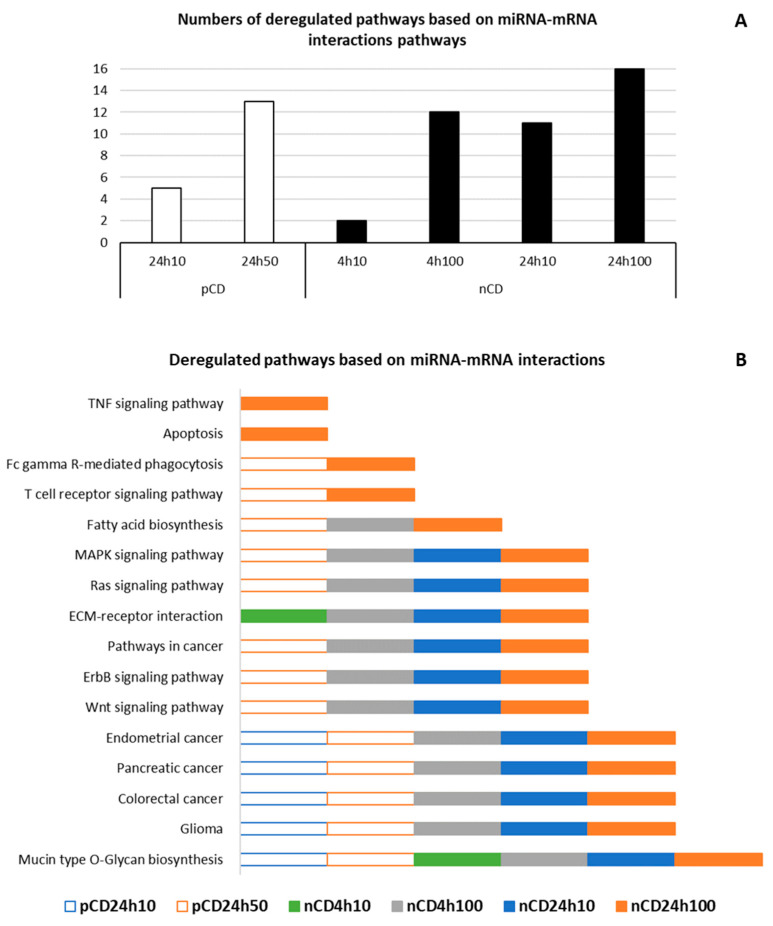
Graphically illustrated overview of deregulated pathways based on miRNA-mRNA interactions after CD NPs exposure. Number (**A**) and identity (**B**) of deregulated pathways per CD, dose and exposure time (empty rectangles denote pCD exposure, filled rectangles denote nCD exposure).

**Table 1 ijms-21-04763-t001:** Selected pathways deregulated after pCD NPs exposure.

ID	Source	Deregulated Pathway	*q*-Value	Deregulated Genes (N)	Genes in Pathway (N)
**pCD 24 h 50 µg/mL**					
83039	KEGG	DNA Replication	<0.001	14	36
1269741	REACTOME	Cell Cycle	<0.001	106	624
1269810	REACTOME	M Phase	<0.001	48	311
1269768	REACTOME	G1/S Transition	<0.001	25	121
1269772	REACTOME	G1/S-Specific Transcription	<0.001	12	18
1269784	REACTOME	DNA strand elongation	<0.001	17	32
1269742	REACTOME	Cell Cycle Checkpoints	<0.001	29	204
1269777	REACTOME	S Phase	<0.001	29	132
1270037	REACTOME	Cholesterol biosynthesis	<0.001	9	24

**Table 2 ijms-21-04763-t002:** Selected pathways deregulated after nCD NPs exposure.

ID	Source	Deregulated Pathway	*q*-Value	Deregulated Genes (N)	Genes in Pathway (N)
**nCD 4 h 100 µg/mL**					
83105	KEGG	Pathways in cancer	<0.001	82	395
1268690	REACTOME	Eukaryotic Translation Elongation	<0.001	49	98
1268679	REACTOME	Eukaryotic Translation Initiation	<0.001	55	127
1268692	REACTOME	Eukaryotic Translation Termination	<0.001	47	97
1269649	REACTOME	Gene Expression	<0.001	345	1844
1268678	REACTOME	Translation	<0.001	68	165
**nCD 24 h 10 µg/mL**					
83039	KEGG	DNA Replication	<0.001	8	36
83055	KEGG	p53 signaling pathway	<0.001	9	69
1269741	REACTOME	Cell Cycle	<0.001	63	624
1269810	REACTOME	M Phase	<0.001	26	311
1269768	REACTOME	G1/S Transition	<0.001	18	121
1269772	REACTOME	G1/S-Specific Transcription	<0.001	5	18
1269784	REACTOME	DNA strand elongation	<0.001	11	32
1269742	REACTOME	Cell Cycle Checkpoints	<0.001	16	204
1269777	REACTOME	S Phase	<0.001	21	132
**nCD 24 h 100 µg/mL**					
83039	KEGG	DNA Replication	<0.001	24	36
1269741	REACTOME	Cell Cycle	<0.001	216	624
1269810	REACTOME	M Phase	<0.001	104	311
1269768	REACTOME	G1/S Transition	<0.001	49	121
1269772	REACTOME	G1/S-Specific Transcription	<0.001	15	18
1269784	REACTOME	DNA strand elongation	<0.001	25	32
1269742	REACTOME	Cell Cycle Checkpoints	<0.001	66	204
1269777	REACTOME	S Phase	<0.001	52	132
1268690	REACTOME	Eukaryotic Translation Elongation	<0.001	38	98
1268679	REACTOME	Eukaryotic Translation Initiation	<0.001	43	127
1268692	REACTOME	Eukaryotic Translation Termination	<0.001	36	97
1270350	REACTOME	DNA Repair	<0.001	80	319

**Table 3 ijms-21-04763-t003:** Selected pathways based on miRNA-mRNA interactions deregulated after pCD NPs exposure.

ID	Deregulated Pathway	*q*-Value	miRNA (N)	Genes (N)
**pCD 24 h 10 µg/mL**				
hsa00512	Mucin type O-Glycan biosynthesis	<0.001	5	11
hsa05214	Glioma	0.049	5	15
hsa05210	Colorectal cancer	0.003	5	20
hsa05212	Pancreatic cancer	0.013	5	19
hsa05213	Endometrial cancer	0.028	6	17
**pCD 24 h 50 µg/mL**				
hsa00512	Mucin type O-Glycan biosynthesis	<0.001	24	26
hsa05214	Glioma	0.005	41	45
hsa05210	Colorectal cancer	<0.001	41	48
hsa05212	Pancreatic cancer	<0.001	41	52
hsa05213	Endometrial cancer	<0.001	41	42
hsa04310	Wnt signaling pathway	<0.001	44	99
hsa04012	ErbB signaling pathway	<0.001	41	66
hsa05200	Pathways in cancer	<0.001	47	266
hsa04014	Ras signaling pathway	<0.001	44	156
hsa04010	MAPK signaling pathway	0.040	44	160
hsa00061	Fatty acid biosynthesis	<0.001	15	8
hsa04660	T cell receptor signaling pathway	0.024	41	71
hsa04666	Fc gamma R-mediated phagocytosis	0.007	41	64

**Table 4 ijms-21-04763-t004:** Selected pathways based on miRNA-mRNA interactions deregulated after nCD NPs exposure.

ID	Deregulated Pathway	*q*-Value	miRNA (N)	Genes (N)
**nCD 4 h 10 µg/mL**				
hsa00512	Mucin type O-Glycan biosynthesis	0.009	9	7
hsa04512	ECM-receptor interaction	<0.001	10	17
**nCD 4 h 100 µg/mL**				
hsa00512	Mucin type O-Glycan biosynthesis	<0.001	46	23
hsa05214	Glioma	<0.001	78	52
hsa05210	Colorectal cancer	0.013	73	50
hsa05212	Pancreatic cancer	0.003	79	53
hsa05213	Endometrial cancer	0.022	72	42
hsa04310	Wnt signaling pathway	0.003	89	108
hsa04012	ErbB signaling pathway	<0.001	86	73
hsa05200	Pathways in cancer	<0.001	94	303
hsa04014	Ras signaling pathway	<0.001	91	175
hsa04010	MAPK signaling pathway	0.018	93	184
hsa04512	ECM-receptor interaction	<0.001	21	27
hsa00061	Fatty acid biosynthesis	<0.001	34	10
**nCD 24 h 10 µg/mL**				
hsa00512	Mucin type O-Glycan biosynthesis	<0.001	26	20
hsa05214	Glioma	<0.001	46	49
hsa05210	Colorectal cancer	0.001	40	46
hsa05212	Pancreatic cancer	<0.001	45	50
hsa05213	Endometrial cancer	0.006	40	40
hsa04310	Wnt signaling pathway	0.007	52	93
hsa04012	ErbB signaling pathway	<0.001	51	66
hsa05200	Pathways in cancer	<0.001	59	260
hsa04014	Ras signaling pathway	0.002	55	145
hsa04010	MAPK signaling pathway	0.004	53	164
hsa04512	ECM-receptor interaction	<0.001	49	55
**nCD 24 h 100 µg/mL**				
hsa00512	Mucin type O-Glycan biosynthesis	<0.001	51	23
hsa05214	Glioma	<0.001	69	55
hsa05210	Colorectal cancer	<0.001	71	57
hsa05212	Pancreatic cancer	<0.001	72	58
hsa05213	Endometrial cancer	<0.001	68	45
hsa04310	Wnt signaling pathway	<0.001	84	108
hsa04012	ErbB signaling pathway	<0.001	80	80
hsa05200	Pathways in cancer	<0.001	91	288
hsa04014	Ras signaling pathway	<0.001	87	171
hsa04010	MAPK signaling pathway	0.002	87	191
hsa04512	ECM-receptor interaction	0.002	72	61
hsa00061	Fatty acid biosynthesis	<0.001	34	9
hsa04660	T cell receptor signaling pathway	<0.001	72	85
hsa04666	Fc gamma R-mediated phagocytosis	0.024	77	66
hsa04210	Apoptosis	0.036	70	66
hsa04668	TNF signaling pathway	0.016	76	84

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
