# Peer review of "The Differential Effect of Carbon Dots on Gene Expression and DNA Methylation of Human Embryonic Lung Fibroblasts as a Function of Surface Charge and Dose"

_ijms, 2020, doi:10.3390/ijms21134763_

Round 1

Reviewer 1 Report

This manuscript reports on a study where the effect of carbon dots, CD, with two different surface charges, likely stemming from surface amino and carboxyl groups, on cell toxicity transcriptomics, miRNA and DNA methylation level were studied. Interestingly, although positive, pCDs were more toxic, their effect on gene expression was smaller than that of negative, nCDs.

Questions

1. It is clear that at a certain concentration, every material or chemical becomes toxic, or induces a change in gene expression and therefore, the choice of exposure concentration of CDs in transcriptomics experiments is crucial. As can be seen from Figure 2, pCDs are more toxic than nCD-s. Can the authors claim that the concentrations of both CD-s used in transcriptomics assay exhibited similar level of "general" toxicity (live-dead analysis)? 

2. Do the authors have an expanation for lower number of differently expressed genes in case of pCD, compared with nCD? This difference varies between ∼10-100-fold, thus, being significant.

3. To compare different transcriptomics results between the different exposures, the authors have chosen to show separately 4h and 24 h (Figure 8). Did the authors also prepare a comparative figure for the different exposure times and concentrations for each particle? 

4. Could the authors relate their results to any of the previous studies on transcriptomics after cellular exposure to CD particles, 10 nm particles, or particles with carboxyl surface groups or amine surface groups. This comparison would enable to reveal the potential mechanism of action of CD particles. Do the authors think that transcriptomics response to CD is CD-specific, particle specific or particle surface specific?  

A comment

As this study is showing just in vitro effects of CD-s, the authors should take care not to make too strong claims. E.g., the last sentence of the abstract "In summary, although the tested CDs induced distinct responses on the level of mRNA and miRNA expression, pathway analyses revealed a common biological impact of both NMs independent of their surface charge" it should be claimed that potential common biological impacts/modes of action were revealed.

Several small mistakes and typos were detected and structures of the sentences need to be revised.

Author Response

Reviewer 1

This manuscript reports on a study where the effect of carbon dots, CD, with two different surface charges, likely stemming from surface amino and carboxyl groups, on cell toxicity transcriptomics, miRNA and DNA methylation level were studied. Interestingly, although positive, pCDs were more toxic, their effect on gene expression was smaller than that of negative, nCDs.

Questions

  1. It is clear that at a certain concentration, every material or chemical becomes toxic, or induces a change in gene expression and therefore, the choice of exposure concentration of CDs in transcriptomics experiments is crucial. As can be seen from Figure 2, pCDs are more toxic than nCD-s. Can the authors claim that the concentrations of both CD-s used in transcriptomics assay exhibited similar level of "general" toxicity (live-dead analysis)?

Response: For the lower tested concentration (10 µg/ml), the general toxicity based on the live-dead analysis is comparable. However, at the higher tested dose pCD exhibited greater toxicity than nCD, even though pCD were tested at 50 µg/ml and nCD at 100 µg/ml. Thus, for the comparison of biological effects of the two nanomaterials the lower tested dose is more relevant. However, it is interesting to note that at this dose few common deregulated mRNAs or miRNAs were detected for either exposure period thus underlining possible differences in biological effects of the tested CD. Another possibility is that although toxicity was not detected using the live-dead assay, there are some inhibitory effects of pCD on gene expression even at the lower tested dose that were manifested by the lower number of deregulated genes after exposure to this nanomaterial (see also response to question 2).

  1. Do the authors have an expanation for lower number of differently expressed genes in case of pCD, compared with nCD? This difference varies between ∼10-100-fold, thus, being significant.

Response: This is a very interesting question. However, it is difficult to explain this observation based on the available data. We speculate that pCD caused some toxic effects even at the lower dose that was not detectable by the live-dead analysis. This potential toxicity may have inhibitory effects on gene expression in general, resulting in lower numbers of deregulated genes observed in our study. We mention this explanation in Discussion (4th paragraph).

  1. To compare different transcriptomics results between the different exposures, the authors have chosen to show separately 4h and 24 h (Figure 8). Did the authors also prepare a comparative figure for the different exposure times and concentrations for each particle?

Response: Yes, we did such comparisons, they are reported in Figure 3. The results showed that for pCD most of the deregulated transcripts were unique for the dose/time of exposure. For nCD, we found some overlap between individual tested concentrations and/or time of exposure. However, only two transcripts were common for all tested conditions.

  1. Could the authors relate their results to any of the previous studies on transcriptomics after cellular exposure to CD particles, 10 nm particles, or particles with carboxyl surface groups or amine surface groups. This comparison would enable to reveal the potential mechanism of action of CD particles. Do the authors think that transcriptomics response to CD is CD-specific, particle specific or particle surface specific?

Response: There is very limited information on transcriptomics response after exposure to CD particles. This was one of the reasons why we decided to conduct such complex genomic analysis in human lung fibroblasts. There is a study performed in plants ([1], also cited in our manuscript) but because of the differences in the model organisms, this data cannot be compared with our results. Even for other nanoparticles, such studies are not common. The effect of silver NP was investigated in intestinal epithelium coculture [2], the impact of poly(D,L)-lactide-co-glycolide NP was assessed in a retinoblastoma model [3]. Also, the response of human primary olfactory cells exposed to ZnO NP with different surface coatings was investigated [4], as well as molecular response of alveolar epithelial cells A549 after exposure to TiO2 NP [5], or boron carbide NP effects in human primary alveolar epithelial cells [6,7]. A combination of diverse nanoparticles and model systems does not allow making general conclusions on transcriptomics effects of NP. It is thus very difficult to identify factor that specifically modulates transcriptomics response induced by CD. Although in our study, surface charge seems to play a significant role (as stated in Discussion, the first and the last paragraph), most probably, it is a combination of physico-chemical properties of NP, exposure system, tested dose and time of exposure.

A comment

As this study is showing just in vitro effects of CD-s, the authors should take care not to make too strong claims. E.g., the last sentence of the abstract "In summary, although the tested CDs induced distinct responses on the level of mRNA and miRNA expression, pathway analyses revealed a common biological impact of both NMs independent of their surface charge" it should be claimed that potential common biological impacts/modes of action were revealed.

Response: We agree with the Reviewer that the results should be interpreted with caution. The statement in the abstract was modified.

Several small mistakes and typos were detected and structures of the sentences need to be revised.

Response: The manuscript was checked by a native English speaker.

References:

[1]          Chen J, Liu B, Yang Z, et al. Phenotypic, transcriptional, physiological and metabolic responses to carbon nanodot exposure in Arabidopsis thaliana (L.). Environ Sci: Nano 2018;5:2672–2685.

[2]          Bouwmeester H, Poortman J, Peters RJ, et al. Characterization of Translocation of Silver Nanoparticles and Effects on Whole-Genome Gene Expression Using an In Vitro Intestinal Epithelium Coculture Model. ACS Nano 2011;5:4091–4103.

[3]          Mitra M, Mohanty C, Harilal A, et al. A novel in vitro three-dimensional retinoblastoma model for evaluating chemotherapeutic drugs. Mol Vis 2012;18:1361–1378.

[4]          Osmond-McLeod MJ, Osmond RI, Oytam Y, et al. Surface coatings of ZnO nanoparticles mitigate differentially a host of transcriptional, protein and signalling responses in primary human olfactory cells. Part Fibre Toxicol 2013;10:54.

[5]          Armand L, Biola-Clier M, Bobyk L, et al. Molecular responses of alveolar epithelial A549 cells to chronic exposure to titanium dioxide nanoparticles: A proteomic view. Journal of Proteomics 2016;134:163–173.

[6]          Türkez H, Arslan ME, Sönmez E, et al. Synthesis, characterization and cytotoxicity of boron nitride nanoparticles: emphasis on toxicogenomics. Cytotechnology 2019;71:351–361.

[7]          Türkez H, Arslan ME, Sönmez E, et al. Microarray assisted toxicological investigations of boron carbide nanoparticles on human primary alveolar epithelial cells. Chemico-Biological Interactions 2019;300:131–137.

[8]          Alley D, Langley-Turnbaugh S, Gordon N, et al. The effect of PM10 on human lung fibroblasts. Toxicol Ind Health 2009;25:111–120.

[9]          Li Y, Wang P, Hu C, et al. Protein corona of airborne nanoscale PM2.5 induces aberrant proliferation of human lung fibroblasts based on a 3D organotypic culture. Sci Rep 2018;8:1939.

[10]        Faber SC, McNabb NA, Ariel P, et al. Exposure Effects Beyond the Epithelial Barrier: Trans-Epithelial Induction of Oxidative Stress by Diesel Exhaust Particulates in Lung Fibroblasts in an Organotypic Human Airway Model. Toxicological Sciences 2020;kfaa085.

[11]        Durantie E, Barosova H, Drasler B, et al. Carbon nanodots: Opportunities and limitations to study their biodistribution at the human lung epithelial tissue barrier. Biointerphases 2018;13:06D404.

[12]        Malina T, Poláková K, Skopalík J, et al. Carbon dots for in vivo fluorescence imaging of adipose tissue-derived mesenchymal stromal cells. Carbon 2019;152:434–443.

Reviewer 2 Report

Authors described that different surface charge of carbon nanodot affected induced different expression pattern of miRNA and mRNA, but not similar biological impacts using one fibroblast cell line. Major and minor revises are as follows.

Major revises

  1. Why authors select fibroblast as cells exposed to nanoparticles? Basically, it is alveolar macrophages and alveolar epithelial cells that phagocytose nanoparticles, not fibroblasts. The function of fibroblasts is considered to change due to signals such as cytokines from alveolar epithelial cells and macrophages exposed to nanoparticles. Please explain the significance of fibroblasts as cells exposed to nanoparticles in the present study.
  2. It is necessary to show more physico-chemical properties of CDs. I would like to confirm that there is not much difference in properties other than surface charge. Please show photograph of CDs in scanning or transmission electron microscope, and measure the primary and secondary diameter of CDs.
  3. It is necessary to investigate whether the nanoparticles are inside or outside the cell. Authors need to investigate the interaction between cells and particles with a confocal microscope.
  4. Will other fibroblasts behave similarly? It is necessary that authors perform this in vitro study using other fibroblasts cell line.
  5. Authors used the word “deregulation” in the manuscript. However the experimental design is not suitable for examining the deregulation of gene expression. In the present study, gene expression is under the stimulation of nanoparticles, and regulated by nanoparticles. Therefore, it is necessary to investigate that gene expression continues to change for 2 weeks (at least) after removing nanoparticles from cultured cells.

Minor revises

  1. Do authors think that these gene expression are a CD specific response, or a common response of nanoparticles, or other? Authors should explain how to consider the gene expression pattern in the present study in discussion.
  2. The exposure concentration of CD in the present study is 10-100 ug/mL. Is this concentration possible in the human environment? Please explain the consideration on exposure concentration considering human environment.
  3. Is the change in gene expression exposed to CD, related to the change in fibroblast phenotype? myofibroblast? Investigating the phenotype of fibroblast by immunostaining (a-SMA antibody) may be a reference for change of gene expression.
  4. How do authors think that the number of change of gene expression exposed to pCD is low in spite of high toxic nanoparticle compared with nCD in shown in Fig 2? However, nCD affect apoptosis.

Author Response

Reviewer 3

Authors described that different surface charge of carbon nanodot affected induced different expression pattern of miRNA and mRNA, but not similar biological impacts using one fibroblast cell line. Major and minor revises are as follows. 

Major revises

  1. Why authors select fibroblast as cells exposed to nanoparticles? Basically, it is alveolar macrophages and alveolar epithelial cells that phagocytose nanoparticles, not fibroblasts. The function of fibroblasts is considered to change due to signals such as cytokines from alveolar epithelial cells and macrophages exposed to nanoparticles. Please explain the significance of fibroblasts as cells exposed to nanoparticles in the present study.

Response: We agree with the Reviewer that lung fibroblasts are not the primary target of nanoparticles (NP) exposure. However, these cells can be affected secondarily, particularly in damaged areas of the lungs. In such tissue, fibroblasts divide and consequently play a major role in the response to inhalation exposure [8,9]. Moreover, they respond to various stimuli, including oxidative stress induced in the epithelial layers of the lungs [10]. It should be also noted that lung fibroblasts represent an established experimental model in genetic toxicology and nanotoxicology. Finally, due to the small size, CDs can easily penetrate the epithelium layer and affect other cells even if the epithelial cells are intact [11].

Apart of fibroblasts, we also performed the identical exposures using human alveolar epithelial cells (A549), however, due to the amount of data we did not include the results to the current manuscript. Although this model is more relevant in terms of biological significance, the cells are of tumor origin and thus the results of gene expression analyses could be distorted by expansion of specific clones and/or chromosomal instability of this cell line.

We included explanation why lung fibroblasts were used in our study to the first paragraph of Discussion.

  1. It is necessary to show more physico-chemical properties of CDs. I would like to confirm that there is not much difference in properties other than surface charge. Please show photograph of CDs in scanning or transmission electron microscope, and measure the primary and secondary diameter of CDs.

Response: Thank you for this important question. In this study we worked with CDs of similar size, that for both nCDs and pCDs is very small with a narrow size distribution of approximately 3 nm and particles having globular shape. Therefore, there should be no influence on resulting toxicity caused by different size of nCDs and pCDs. We measured also hydrodynamic diameter (DLS) of both CDs and we observed similar sizes where the maximum intensity was 5 nm for pCDs and 6 nm for nCDs.

Information about the size and shape is added in the revised manuscript in Results (1st paragraph) and illustrated by new figures in Supplementary Materials (Figure S1). We also added a description of TEM and DLS analyses to the Materials and Methods, section (4.1).

  1. It is necessary to investigate whether the nanoparticles are inside or outside the cell. Authors need to investigate the interaction between cells and particles with a confocal microscope.

Response: We performed confocal microscopy after 24 hours of incubation in order to see the uptake of both nanoparticles in detail. The results are now reported in Figure S2 in which Z-stack of four different layers of selected cells is showed. After 24 h of incubation and washing, both CDs are located inside the cells. Thus, both types of nanoparticles were taken up by HEL12469 cells and internalized in vesicles.

The results are described in section 2.2; methodological aspects of confocal microscopy Materials and Methods (section 4.1).

  1. Will other fibroblasts behave similarly? It is necessary that authors perform this in vitro study using other fibroblasts cell line.

Response:  As mentioned above, we performed identical study using another cell line – human alveolar epithelial cells (A549 cells). We believe, that from the biological point of view it is more interesting to compare the response of cells of different origin, rather than repeating the experiments with the cells of the same type. Due to the amount of data, the results from A549 cells will be included in a separate manuscript. However, to illustrate our observations from this cell line, we attach a file reporting numbers of unique and common deregulated mRNAs and miRNAs in A549 cells treated with pCDs and rCDs. These results correspond to the data in Figure 3, 8, 10 and 11 of our manuscript. Interestingly, although the pattern of distribution of common and unique deregulated genes is similar for both cell lines, the numbers of deregulated genes is about twice as high in lung fibroblasts than in epithelial cells. It indicates differences in biological response after CDs exposure between both cell lines.

If needed, more results obtained for A549 cells can be provided to the Reviewer. However, we feel that adding additional data to the manuscript would make the text too complex.

  1. Authors used the word “deregulation” in the manuscript. However the experimental design is not suitable for examining the deregulation of gene expression. In the present study, gene expression is under the stimulation of nanoparticles, and regulated by nanoparticles. Therefore, it is necessary to investigate that gene expression continues to change for 2 weeks (at least) after removing nanoparticles from cultured cells.

Response: In our study, we used a standard approach established in genetic toxicology. We agree that “recovery” tests may also yield interesting data. However, they do not reflect the results of acute exposure. Concerning gene expression, it usually changes dynamically and quite rapidly. Thus, it is probable that after 2 or more weeks no gene expression changes would be detectable. Moreover, the cell cultures could not be grown for such a long period of time without passaging, which would further “dilute” the cells affected by the primary exposure and decrease a chance of any positive observation.

Minor revises

  1. Do authors think that these gene expression are a CD specific response, or a common response of nanoparticles, or other? Authors should explain how to consider the gene expression pattern in the present study in discussion.

Response: We answered a similar question in Reviewer’s 1 comments. Based on our data, it is rather difficult to identify factors that specifically modulate transcriptomics response induced by CD. However, we believe that in our study, surface charge plays a major role in this response. This fact is mentioned in Discussion (the first and the last paragraph). In addition, it is probable that a combination of physico-chemical properties of NP, exposure system, tested dose and time of exposure are other factors affecting transcriptomics response.

  1. The exposure concentration of CD in the present study is 10-100 ug/mL. Is this concentration possible in the human environment? Please explain the consideration on exposure concentration considering human environment.

Response: The exposure to CDs is related, among others, to medical applications, including stem cells tracking. In a recent study, a dose of 100 µg/ml was used for mesenchymal stromal cells labeling. The cells were then visualized in experimental mice [12]. Thus, even though CDs have not been applied in humans yet, their potential application in medicine was done at a dose corresponding to treatments in used in our study.

  1. Is the change in gene expression exposed to CD, related to the change in fibroblast phenotype? myofibroblast? Investigating the phenotype of fibroblast by immunostaining (a-SMA antibody) may be a reference for change of gene expression.

Response: We believe that the exposure is too short (4h and 24h) to induce changes resulting in differentiation into myofibroblasts. Although we did not perform immunostaining, we regularly observed the cells under a microscope and did not detect any morphological changes.

  1. How do authors think that the number of change of gene expression exposed to pCD is low in spite of high toxic nanoparticle compared with nCD in shown in Fig 2? However, nCD affect apoptosis.

Response: As mentioned in a response to Reviewer 1, we speculate that pCD cause some toxic effects even at the low tested concentration (10 µg/ml) at which no significant changes were observed using live-dead analysis. This toxicity may be associated with inhibitory effects on gene expression causing deregulation of a lower number of deregulated genes in cell treated with pCD. This information was added to Discussion, 4th paragraph.

References:

[8]          Alley D, Langley-Turnbaugh S, Gordon N, et al. The effect of PM10 on human lung fibroblasts. Toxicol Ind Health 2009;25:111–120.

[9]          Li Y, Wang P, Hu C, et al. Protein corona of airborne nanoscale PM2.5 induces aberrant proliferation of human lung fibroblasts based on a 3D organotypic culture. Sci Rep 2018;8:1939.

[10]        Faber SC, McNabb NA, Ariel P, et al. Exposure Effects Beyond the Epithelial Barrier: Trans-Epithelial Induction of Oxidative Stress by Diesel Exhaust Particulates in Lung Fibroblasts in an Organotypic Human Airway Model. Toxicological Sciences 2020;kfaa085.

[11]        Durantie E, Barosova H, Drasler B, et al. Carbon nanodots: Opportunities and limitations to study their biodistribution at the human lung epithelial tissue barrier. Biointerphases 2018;13:06D404.

[12]        Malina T, Poláková K, Skopalík J, et al. Carbon dots for in vivo fluorescence imaging of adipose tissue-derived mesenchymal stromal cells. Carbon 2019;152:434–443.

Round 2

Reviewer 1 Report

The authors have responded to my previous questions. Although the manuscript could have benefited from more comprehensive explanation of the potential biological consequences of CD exposure, it is clear that no significant generalizations could be made at that moment, mainly due to the lack of prior comparative studies. 

Reviewer 2 Report

No